# MIPT: Multilevel Informed Prompt Tuning for Robust Molecular Property Prediction

Yeyun Chen [1 2]  Jiangming Shi [1 2]

## Abstract

The progress in materials science and drug discovery is impeded by the availability of labeled data and the high costs of manual annotation, driving the need for efficient strategies to capture molecular representations and enable accurate predictions. Pretrained Graph Neural Networks have shown promise in capturing universal molecular representations, but adapting them to task-specific applications remains challenging. In this paper, we propose Multilevel Informed Prompt-Tuning (MIPT), a novel framework for effectively tailoring pretrained models to molecule-related tasks. MIPT utilizes a lightweight, multi-level prompt learning module to capture node-level and graph-level task-specific knowledge, ensuring adaptable and efficient tuning. Additionally, a noise penalty mechanism is introduced to address mismatches between pretrained representations and downstream tasks, reducing irrelevant or noisy information. Experimental results show that MIPT surpasses all baselines, aligning graph space and task space while achieving significant improvements in molecule-related tasks, demonstrating its scalability and versatility for molecular tasks.

## 1. Introduction

Machine learning (ML) has demonstrated significant potential to revolutionize the discovery and development of new materials (Chibani & Coudert, 2020), drugs (Shen & Nicolaou, 2019), and chemical processes (Taskinen & Yliruusi, 2003). However, the rapid growth in these fields is hindered by the scarcity of labeled molecular data and the high cost of manual annotation. These challenges have driven interest in strategies that efficiently capture molecular properties and enable accurate predictions with minimal supervision. Pretrained Graph Neural Networks (GNNs), designed to learn universal molecular representations, offer a promising solution. Yet, adapting these representations to task-specific applications remains a key challenge, necessitating efficient and flexible tuning methods. The pretrain-and-finetune paradigm (Hu et al., 2019) has been proposed to address this challenge, but the gap between pre-training objectives and downstream tasks often hinders effective knowledge transfer, leading to suboptimal performance.

Prompt-based learning (Sun et al., 2023b; Li et al., 2024) has recently emerged as a powerful method for bridging this semantic gap. By aligning downstream tasks with pre-trained feature spaces, prompts facilitate knowledge transfer, improving accuracy and efficiency in applications such as molecular property prediction. Graph prompts, in particular, have shown efficacy in narrowing this gap. Some approaches use manually customized prompts (Wang et al., 2024), incorporating task-specific substructures or functional groups, while others explore universal prompts, such as GPF (Fang et al., 2024), which are applicable across all nodes, enhancing task adaptability and generalization while reducing computational costs. Additionally, in-context learning methods like PRODIGY (Huang et al., 2024) leverage label nodes to construct prompt graphs, transforming downstream tasks into link prediction problems and enhancing contextual learning. These approaches collectively provide more flexible and efficient solutions for downstream applications.

Despite these advances, significant challenges remain in applying prompts to pre-trained GNNs. Many existing graph prompts are unable to optimize for specific tasks and exhibit limited scalability. While manually designed prompts (Wang et al., 2024) offer task-specific flexibility, their reliance on predefined templates limits adaptability. Similarly, universal prompts, such as GPF-plus (Fang et al., 2024), struggle to capture the diversity of all nodes, and assigning distinct prompts to each node incurs high computational costs (Sun et al.). Some studies address these issues by introducing label nodes to construct graph prompt (Huang et al., 2024), but this approach can disrupt node dependen-

---

[1]Institute of Artificial Intelligence, Xiamen University, Xiamen Fujian, China [2]Shanghai Innovation Institute, Shanghai, China. Correspondence to: Jiangming Shi <jiangming.shi@outlook.com>.

*Proceedings of the 42$^{nd}$ International Conference on Machine Learning*, Vancouver, Canada. PMLR 267, 2025. Copyright 2025 by the author(s).

dencies and alter graph topology, potentially reducing prediction accuracy. Moreover, the large number of tunable parameters in pre-trained GNNs, combined with the limited availability of labeled molecular data, makes downstream fine-tuning less effective. These limitations highlight the need for adaptable, scalable, and efficient prompt designs to fully leverage the potential of pre-trained GNNs.

On the other hand, while prompts are effective in enhancing generalization, they may inadvertently introduce task-irrelevant noise, which can hinder the model's ability to prioritize critical features, ultimately leading to suboptimal performance in specific tasks. For instance, in graph classification, prompts may overemphasize edge or node details while neglecting the overall graph structure. Similarly, pretrained GNNs, despite capturing broad feature distributions, may fail to align with task-specific requirements, such as mismatches between node-level pre-trained features and subgraph-level features needed downstream, causing overfitting and reduced accuracy. Additionally, downstream tasks often exhibit biases, prioritizing high-probability features while ignoring low-probability tail samples. Effectively bridging the gap between generalized pre-trained knowledge and domain-specific needs presents a critical challenge, necessitating highly adaptive approaches to mitigate noise, address feature mismatches, and align with task-specific biases for enhanced performance.

Based on the above observations, this paper aims to address the following challenges: (i) While prompts are designed to bridge the gap between pre-training and downstream tasks, universal prompts frequently exhibit limited intuitive interpretability and poor scalability. Consequently, effective prompt design necessitates integrating graph-specific information, capturing their adaptability to graph structures, and ensuring efficient scalability. (ii) Although prompts are intended to guide models towards task adaptation, their application may inadvertently introduce task-irrelevant noise due to suboptimal feature modeling, thereby compromising performance. Therefore, it is crucial to effectively extract salient features, mitigate noise interference, and enhance the model's robustness and adaptability to downstream tasks.

In this paper, we propose a novel framework, Multilevel Informed Prompt Tuning (**MIPT**), to enhance pretrained GNNs for molecule-related tasks. The framework employs a lightweight multi-level prompt learning network to capture task-specific knowledge at both node and graph levels, enabling effective adaptation to diverse scenarios. Furthermore, a noise penalty mechanism is introduced to align distributions and facilitate knowledge transfer across graph spaces. Experiments on public datasets demonstrate that MIPT achieves superior performance in molecular property prediction while requiring fewer trainable parameters.

Our contributions can be summarized as below:

- We introduced a novel framework called Multilevel Informed Prompt Tuning (MIPT), which leverages a lightweight multi-level prompt learning module to effectively capture task-specific knowledge at both the node and graph levels, enabling efficient adaptation of pretrained GNNs to molecular tasks.

- We incorporates a noise penalty mechanism to address mismatches between pretrained representations and downstream tasks, effectively reducing irrelevant or noisy information and enhancing task performance.

- Experimental results demonstrate that MIPT surpasses baseline model, excelling in aligning graph and task space modalities, while showcasing scalability and versatility across a wide range of molecule-related tasks.

## 2. Related Works

### 2.1. Pre-trained GNNs

Pretraining and fine-tuning are widely used to transfer knowledge from related tasks and enhance model generalization. Pretraining trains on large-scale data via self-supervised or supervised tasks, followed by fine-tuning on smaller labeled datasets. In molecular GNNs, various self-supervised tasks capture chemical rules and patterns at node, subgraph, and graph levels (Xia et al., 2022b). However, pretrained GNNs face challenges in extracting task-relevant knowledge and often suffer from overfitting during fine-tuning (Xia et al., 2022a). Unlike NLP and CV, pretrained GNNs do not consistently improve downstream performance (Sun et al., 2022b), partly due to limited research on selecting effective self-supervised tasks. Most studies adopt only a few tasks for pretraining, resulting in GNN models excelling in specific downstream tasks but lacking consistency (Sun et al., 2022b). Moreover, recent findings suggest self-supervised pretraining sometimes fails to outperform non-pretrained methods (Sun et al., 2022b). Overall, pretrained GNNs offer limited advantages over non-pretrained models. Recent approaches have attempted to improve fine-tuning using regularization (Xuhong et al., 2018) or update constraints (Houlsby et al., 2019; Xia et al., 2022a; Zhang et al., 2022).

### 2.2. Graph Prompt Learning

Despite the advances in pre-trained models, bridging this gap between pre-trained and fine-tuned models has become a critical focus in recent research. GPPT (Sun et al., 2022a) introduced a learnable prompt mechanism for graphs, specifically targeting node classification tasks. While effective in this specific domain, its design lacks generalizability and cannot adapt to other downstream tasks. Building upon this, Graph Prompt (Liu et al., 2023) proposed a novel task-specific learnable prompt, which guides the ReadOut

operation of each downstream task using an appropriate aggregation scheme. However, this approach faces challenges in multitasking scenarios, as differences across tasks make it difficult to identify a universal prompt pattern, ultimately limiting its effectiveness. All in One (Sun et al., 2023a) addressed the issue of multitasking by designing a unified training framework for joint multi-task prompts. While this approach leverages meta-learning to accommodate a large variety of tasks, it is computationally expensive and exhibits limited generalization capabilities across diverse tasks. GPF and its extended version, GPF-plus (Fang et al., 2024), introduced learnable prompts at the node level, aiming for versatility across different downstream tasks. This approach effectively balances adaptability and computational efficiency, presenting a more practical solution for a broad range of task requirements.

## 3. Preliminaries

### 3.1. Problem Formulation

**Notations.** Let $\mathcal{G} = (\mathcal{V}, \mathcal{E}) \in \mathbb{G}$ denotes a graph, where $\mathcal{V} = \{v_1, ..., v_N\}$ represents the set of nodes, and $\mathcal{E} \subseteq \mathcal{V} \times \mathcal{V}$ represents the set of edges. $\mathcal{G}$ is associated with a node feature matrix $\mathbf{X} \in \mathbb{R}^{N \times F}$ where $F$ is the feature dimension, and an adjacency matrix $\mathbf{A} \in \mathbb{R}^{N \times N}$ where $\mathbf{A}_{ij} = 1$ if and only if $(v_i, v_j) \in \mathcal{E}$ and $\mathbf{A}_{ij} = 0$ otherwise.

**Problem Definition.** Given a pre-trained GNN $f$, a learnable projection layer $g$, and a downstream task dataset $D = \{(\mathcal{G}_1, y_1), (\mathcal{G}_2, y_2), \ldots, (\mathcal{G}_m, y_m)\}$, our objective is to fine-tune the parameters of the prompt $p_g$ and the projection layer $g$ to effectively bridge the gap between the pre-trained GNN $f$ and the downstream task. Specifically, the aim is to maximize the likelihood of accurately predicting the ground truth labels $y$ associated with the downstream task, formulated as follows:

$$\max_{p_g, g} P_{p_g, g}(y \mid \mathcal{G}) \tag{1}$$

### 3.2. Molecule Representation Learning.

We adopt the Graph Isomorphism Network (GIN) (Xu et al., 2018) as the backbone for our models, leveraging its strong expressive power and effectiveness in graph-related tasks. In fact, our framework is model-agnostic, enabling seamless integration with various existing molecular prediction model as downstream models. The pre-training process is designed to learn general-purpose graph representations by optimizing auxiliary objectives on large-scale, unlabeled graph data. This process primarily follows two paradigms: self-supervised learning and contrastive learning (Zhou et al., 2024).

Consider a molecule graph $\mathcal{G} = (\mathcal{V}, \mathcal{E})$. Typically, the predictor $\rho$ can be expressed as $f \circ g$, comprising two compo-

nents: a GNN encoder $f: \mathcal{G} \to \mathbb{R}^d$, which generates molecular representations, and a projection layer $g : \mathbb{R}^d \to \mathcal{Y}$. In this context, the projection layer serves as a downstream classifier, predicting labels from the generated representations. Specifically, the encoder $f$ operates in two stages. The first stage utilizes a GNN to learn node-level representations $h_v^{(k)}$ as formulated as follows:

$$\mathbf{h}_v^{(k)} = \phi^{(k)}\left(\mathbf{h}_v^{(k-1)}, \varphi^{(k)}\left(\left\{\mathbf{h}_u^{(k-1)} : u \in \mathcal{N}(v)\right\}\right)\right) \tag{2}$$

where $\mathbf{h}_v^{(k)}$ represents the node embedding of node $v$ at the $k$-th layer, $\mathcal{N}(v)$ is the set of neighbors of node $v$, $\varphi^{(k)}(\cdot)$ is the aggregation function at the $k$-th iteration, and $\phi^{(k)}(\cdot)$ is the combination function at the $k$-th iteration. The node embeddings are initialized as $\mathbf{h}_v^{(0)} = \mathbf{X}_v$, where $\mathbf{X}_v$ denotes the input features of node $v$.

To ensure effective feature updates, the feature transformation at layer $k$ can be expressed as:

$$\mathbf{h}_v^{(k)} = \text{MLP}^{(k)}\left(\left(1 + \epsilon^{(k)}\right)\mathbf{h}_v^{(k-1)} + \sum_{u \in \mathcal{N}[v]} \mathbf{h}_u^{(k-1)}\right) \tag{3}$$

where $\epsilon^{(k)}$ is a learnable parameter, $\text{MLP}^{(k)}$ represents a multi-layer perceptron applied at the $k$-th layer, and $\mathcal{N}[v]$ includes the neighbors of node $v$ along with the node itself.

Secondly, at the final layer $K$, the GNN aggregates the learned node embeddings using a READOUT function:

$$\mathbf{h}_v = \text{READOUT}(\{\mathbf{h}_v^K : v \in \mathcal{V}\}) \tag{4}$$

where READOUT can be implemented as a sum, mean, or maximum over all node embeddings in the graph.

## 4. Method

This section provides a thorough explanation of the proposed MIPT framework, detailing its technical design and the motivations behind its development. MIPT is specifically designed to address two primary challenges observed in current prompt-tuning approaches: (i) Universal prompts often lack the necessary flexibility and interpretability to adapt to diverse task-specific requirements, and (ii) prompts can inadvertently introduce task-irrelevant noise, which can significantly degrade overall model performance.

To tackle these issues, we present a comprehensive overview of the framework's core components: the multi-level graph-informed prompts and denoising prompt mechanisms. The overall architecture is illustrated in Figure 1.

### 4.1. Parameter-efficient Tuning for Node Encoder

**Efficient Tuning with LoRA Update.** To address the gap between pre-training objectives and downstream task requirements, we propose an efficient fine-tuning approach

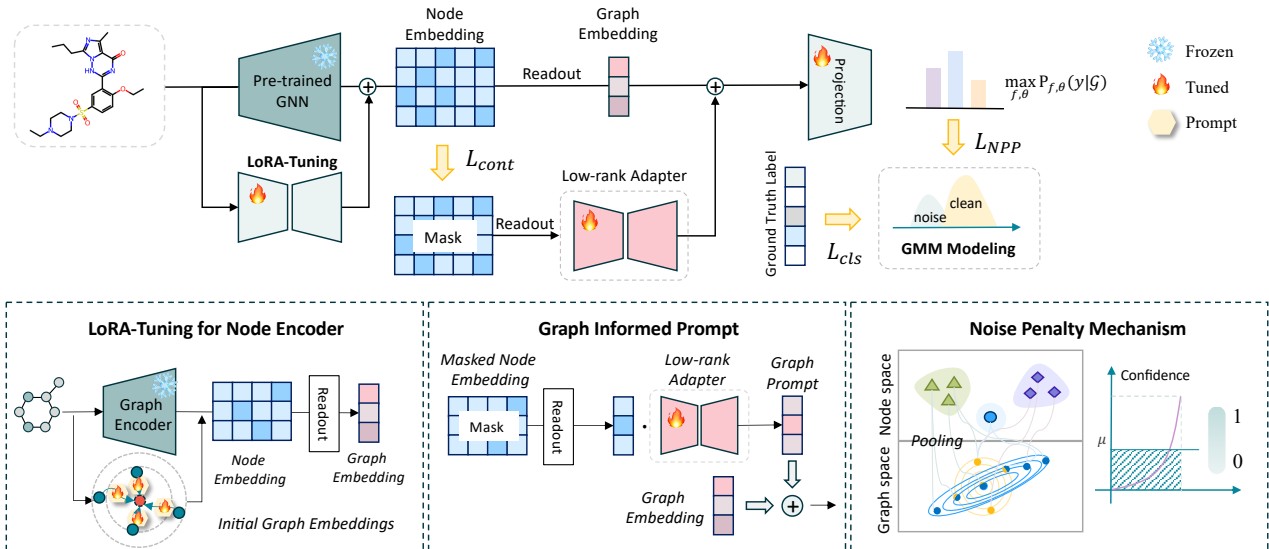

*Figure 1.* Comprehensive overview of the MIPT framework. Our proposed framework synergistically combines efficient node fine-tuning and a unified distribution denoising mechanism, thereby providing an efficient and robust solution for local-global prompt-tuning across varied scenarios.

that integrates a low-rank adaptation matrix for learning task-specific embeddings. This method prioritizes features pertinent to downstream tasks while preserving the expressiveness of the pre-trained model. For the frozen pre-trained GNN weight matrix $W_0 \in \mathbb{R}^{d \times k}$, the parameter update $\Delta W$ is represented using a low-rank decomposition. The feature update process, as described in Eq. 5, is reformulated as:

$$h_v^{(k)} = (W_0^{(k)} + \Delta W^{(k)}) \sum_v \left( \left(1 + \epsilon^{(k)}\right) \mathbf{h}_v^{(k-1)} + \sum_{u \in \mathcal{N}[v]} \mathbf{h}_u^{(k-1)} \right) \tag{5}$$

here, $W_0$ denotes the weight matrix of the pre-trained MLP layer, and $\Delta W$ represents the low-rank adaptation. The parameter $\Delta W$ is further expressed as:

$$\Delta W^{(k)} = B^{(k)} A^{(k)} \tag{6}$$

where $A \in \mathbb{R}^{r \times d}$ and $B \in \mathbb{R}^{d \times r}$ are low-rank matrices with $r \ll d$. The low-rank structure of $\Delta W$ ensures that the additional parameter number is minimal, thus enhancing computational efficiency. These trainable low-rank matrices are injected into the node MLP layer to approximate weight updates effectively.

**Proposition 4.1.** *For a pretrained GNN model $f$, with parameters* $\mathbf{W}$*, node features* $\mathbf{X}$*, there satisfies that incorporating LoRA parameters into the linear transformation during node updates does not alter the underlying graph structure.*

A detailed proof for proposition 4.1 refers to Appendix A.

### 4.2. Multilevel Graph-Informed Prompt

For downstream prediction tasks, obtaining task-specific graph representations $h_g$ is essential, as graph-level embeddings directly influence model performance. However, existing pooling operations fail to satisfy the expressiveness conditions (Bianchi & Lachi, 2023). To address this, we propose Multilevel Graph-Informed Prompt (MGIP), a novel method that extracts features from node-level embeddings and constructs low-rank graph prompt after aggregation. By encoding domain-specific information directly into $h_g$, our approach enhances the adaptability and efficiency of graph representations.

**Random Node Masking.** Let $h_v^* \in \mathbb{R}^{|V| \times d}$ be the node embeddings with LoRA tuning initialized using Kaiming initialization (He et al., 2016). Subsequently, we employ a random masking method to process $h_v^*$, aiming to enhance the model's robustness and its ability to identify critical subgraphs. This method involves three main steps: segmenting, filling, and random masking, ensuring both structural and representational variability are effectively captured. We initialize a binary matrix $\mathbf{M} \in \{0, 1\}^{b \times S_{\max}}$, where $b$ is the batch size and $S_{\max}$ is the maximum number of nodes in the batch. To ensure all samples in the batch share the same input dimensions, we apply a zero-padding operation to the node representations of each sample. The padding is defined as:

$$\mathbf{M}_{i,j} = \begin{cases} 1, & \text{if } j < s_i \\ 0, & \text{otherwise} \end{cases} \tag{7}$$

where $s_i$ represents the size of the $i$-th graph. A random

masking matrix $\mathbf{R} \in \{0, 1\}^{b \times S_{\max}}$ is generated, with each element sampled from a Bernoulli distribution: $\mathbf{R}_{i,j} \sim$ Bernoulli$(1-p)$, where $p$ is the masking probability (e.g., $p = 0.2$). The final mask matrix is updated as: $\mathbf{M} \leftarrow \mathbf{M} \odot \mathbf{R}$, where $\odot$ denotes element-wise multiplication. The updated mask is applied to the node $v$ to produce the masked embeddings $h'_v$:

$$h'_v = \begin{cases} h^*_v, & \text{if } \mathbf{M}_{i,j} = 1 \\ 0, & \text{otherwise} \end{cases} \tag{8}$$

where $j$ in $\mathbf{M}_{i,j}$ is the $i$-th graph of the $j$-th node. By combining deterministic segmentation with random masking, this approach balances structural consistency with variability, encouraging the model to identify key subgraphs and improving generalization. This process ensures the graph representations retain critical information while being robust to structural noise.

**Global-Local Information Maximization.** To obtain the global representation of the entire graph, we utilize a readout function $\Gamma : h'_v \in \mathbb{R}^{|\mathcal{V}| \times d} \to h'_g \in \mathbb{R}^d$ (i.e., $h'_g = \Gamma(h'_v)$, $h_g = \Gamma(h_v)$), which aggregates the local features into a global context. To improve discriminative power, contrastive learning (CL) is introduced to maximize mutual information between original and masked features. The contrastive learning loss function is formulated as:

$$\mathcal{L}_{\text{cont}} = $$
$$-\frac{1}{N} \sum_{i=1}^{N} \log \frac{\exp\left(\varphi\left(\mathrm{h}'_{g,i}, \mathrm{h}_{g,i}\right)\right)}{\exp\left(\varphi\left(\mathrm{h}'_{g,i}, \mathrm{h}_{g,i}\right)\right) + \sum_{j=1, i \neq j}^{N} \exp\left(\varphi\left(\mathrm{h}'_{g,j}, \mathrm{h}_{g,j}\right)\right)} \tag{9}$$

$$\varphi(\cdot) = (h'_{g,i})^\top (h_{g,i})/\tau \tag{10}$$

where $\tau$ is the temperature parameter. CL enhances the correlation between local node representations and global graph representations, improving the multi-level expressiveness of graph embeddings.

**Graph Informed Prompt Generation.** Then, we propose the initialization of graph prompt is guided by graph masked feature. To capture domain-specific graph features, we introduce a lightweight prompt function, denoted as $\Phi(\cdot)$, to provide implicit prompts $p_g$. Formally, this prompt function, is defined as follows:

$$\Phi(h'_g) = W_B(W_A \cdot h'_g) \tag{11}$$

The trainable parameters of this prompt function are denoted as $W_A \in \mathbb{R}^{d' \times r}$ and $W_B \in \mathbb{R}^{r \times d'}$, where $r \ll d'$. The domain-specific prompt is combined with the graph features from pretrained encoder, creating rich information that enables the effective identification of context-specific changes in critical structures. This, in turn, facilitates precise modeling of structural correlations.

**Task-specific Projection Head.** After that, the refined graph representation, $h_G = \text{CONCAT}(h_g, p_g)$, is subsequently passed through a projection head to produce the

final output:

$$\hat{y} = g(h_G) \tag{12}$$

where $g$ is the prediction function. This modification enables downstream models to efficiently leverage the prompts $p_g$ for adaptation to new data and tasks.

### 4.3. Optimization

In the prompt-tuning phase, we specifically update the parameters of the prompt and downstream by performing a limited training on an unseen dataset. This approach enables downstream models to efficiently adapt to new data.

**Training Procedures.** To train the model, a task-specific projection head is used.

$$\mathcal{L}^i_{cls} = \log P\left(y^i \mid g\left(h_G\right)\right) \tag{13}$$

For graph classification, the loss is typically the binary cross-entropy loss:

$$\mathcal{L}_{cls}(g_\theta) = -\sum_{i=1}^{N} \left(y_i \log(\hat{y}_i) + (1 - y_i) \log(1 - \hat{y}_i)\right) \tag{14}$$

where N is the number of training samples, $y_i$ is the true label, and $\hat{y}_i$ is the predicted label of i-th sample.

**Noise Penalty Mechanism.** Building on Arpit et al. (2017), it has been demonstrated that deep neural networks (DNNs) initially learn simpler and more generalizable patterns during the early stages of training, before gradually overfitting to noisy or degenerate samples. Inspired by this observation, we model the confidence distribution of samples based on their classification loss (Shi et al., 2024). Specifically, we employ a two-component Gaussian Mixture Model (GMM) to represent the loss distribution derived from Eq. (13):

$$p(\mathcal{L}_{cls} \mid \theta) = \sum_{t=1}^{T} \pi_t \Phi(\mathcal{L}_{cls} \mid t) \tag{15}$$

where $h_G$ is pretrained molecular features, $g(\cdot)$ is downstream task with parameters $\theta$, and $\pi_t$ and $\Phi(\mathcal{L}_{cls} \mid t)$ represent the mixture coefficient and the probability density of the $t$-th Gaussian component, respectively.

The GMM, $\Phi(\mathcal{L}_{cls} \mid t) \to p$, is utilized to model the posterior probability, denoted as $w_i(0 \leq w_i \leq 1)$, which is computed to quantify the confidence of the $i$-th sample:

$$w_i = p\left(t \mid \mathcal{L}^i_{cls}\right) \tag{16}$$

where k is the Gaussian component with the smaller mean, and $p\left(t \mid \mathcal{L}^i_{cls}\right)$ represents the posterior probability of $\mathcal{L}^i_{cls}$ under the $t$-th component. The confidence $\hat{w}_i$ is then updated as follows:

$$\hat{w}_i = \begin{cases} 1, & \text{if } w_i > \mu \\ 0, & \text{otherwise} \end{cases} \tag{17}$$

Here, $\mu$ is the average of $w_i$, i.e. mean$(w_i)$. With the confidence $\hat{w}$ effectively capturing high-relevant representation, we introduce the Noise Prompts Penalty (NPP) loss, $\mathcal{L}_{NPP}$, to suppress the influence of redundancy noise features. The loss is defined as:

$$\mathcal{L}_{NPP} = -\frac{1}{N} \sum_{i=1}^{N} \hat{w}_i \log P\left(y^i \mid g\left(h_G\right)\right) \qquad (18)$$

This mechanism leverages $\hat{w}$ to penalize noisy features, ensuring that the model focuses on clean and reliable representations, thereby improving its robustness and overall performance.

**Final Objectives.** The overall training objective for the downstream task is illustrated as follows:

$$\mathcal{L}_{total} = \mathcal{L}_{NPP} + \mathcal{L}_{cont} \qquad (19)$$

The model parameters $\theta$ are updated by minimizing $\mathcal{L}_{total}$ using gradient-based optimization.

Compute the gradient of $\mathcal{L}_{total}$ with respect to the model parameters:

$$\nabla_\theta \mathcal{L}_{total} = \nabla_\theta \mathcal{L}_{NPP} + \nabla_\theta \mathcal{L}_{cont} \qquad (20)$$

$$\theta \leftarrow \theta - \eta \nabla_\theta \mathcal{L}_{total}, \qquad (21)$$

where $\eta$ is the learning rate. The parameters of GMM are updated in alternating steps using the Expectation-Maximization (EM) algorithm, with the pseudocode provided in the Appendix 1.

The training process alternates between updating the model parameters $\theta$ and the GMM parameters until convergence. By progressively refining both components, the model learns to focus on reliable samples, while reducing the impact of noisy prompts, ultimately improving the robustness and accuracy of predictions.

**Proposition 4.2.** *(Robustness of NPP) Given a classification task where prompts $p_g$ are used to adapt a pretrained model $f(\cdot)$, the incorporation of the NPP loss $\mathcal{L}_{NPP}$ improves robustness by selectively reducing the influence of noisy prompt.*

A detailed proof for proposition 4.2 is given in Appendix B.1.

# 5. Experiments

In this section, we present an extensive set of experiments to evaluate the performance of our model. The analysis is structured around addressing the following key research questions:

- **RQ1:** How does the efficiency of MIPT compare to that of prompt-tuning and end-to-end training approaches?

- **RQ2:** Does MIPT effective in generalizing pre-trained models to new molecular property prediction datasets and tasks?

- **RQ3:** What impact do the key components of MIPT have on the performance of downstream models?

- **RQ4:** How do hyperparameters influence the performance of MIPT on molecular property prediction?

## 5.1. Experiment Setup

**Datasets.** We employ eight common datasets from MoleculeNet (Wu et al., 2018) as our benchmark datasets: BBBP, Tox21, ToxCast, SIDER, ClinTox, MUV, HIV and BACE. Random splits and scaffold splits for these datasets are adopted. More dataset and training details are available in the Appendix C.1.

**Pretraining strategies.** We take into account four pretrained models for comparison: Deep Graph Infomax (Infomax), Attribute Masking (AttrMasking), Context Prediction (ContextPred), and Edge Prediction (EdgePred).

**Baselines.** We consider three tuning strategies for comparison is used to evaluate: FT(Fine-Tuning), GPF (Fang et al., 2024), GPF-plus (Fang et al., 2024). In the case of freezing the pre-trained model, our prompt method aims to learn the input graph and reconstruct the downstream task to fit the pre-training strategy.

A variety of baselines are included for an in-depth comparison. Supervised GNN methods include D-MPNN (Yang et al., 2019), MGCN (Lu et al., 2019), and AttentiveFP (Xiong et al., 2019). Meanwhile, pretraining approaches consist of N-gram (Liu et al., 2019), PretrainGNN (Hu et al., 2019), GROVER (Rong et al., 2020), 3D-Infomax (Stärk et al., 2022), GraphMVP (Liu et al., 2021), MolCLR (Wang et al., 2022), Uni-Mol (Zhou et al., 2023), and InstructMol (Wu et al.).

**Metric.** ROC-AUC scores are evaluated on the test set. We report the mean and standard deviation of the results from three random seeds.

## 5.2. Overall Performance (RQ1)

We evaluated the performance of downstream tasks across various pre-training and tuning strategies, with results summarized in Table 1. Our method consistently outperforms all baselines, demonstrating its effectiveness and robustness. It achieves significant improvements over fine-tuning strategies across all datasets. While universal learnable prompts in baseline approaches yield limited gains and lack consistency, our method achieves the highest ROC-AUC (%) scores on most datasets, as highlighted in bold.

For example, on the BBBP dataset, our method achieves

*Table 1.* Graph classification (ROC-AUC scores %, higher is better ↑) results across 8 datasets. The best results are highlighted in **boldface**, while the second-best results are marked with *underline*.

| Pretrained strategy | Datasets
**# Molecules**
**# Tasks** | **BBBP↑**
2039
1 | **Tox21↑**
7831
12 | **ToxCast↑**
8576
617 | **SIDER↑**
1427
27 | **ClinTox↑**
1477
2 | **MUV↑**
41127
1 | **HIV↑**
93087
17 | **BACE↑**
1513
1 |
|---|---|---|---|---|---|---|---|---|---|
| Infomax | + FT | 67.55(2.06) | 78.57(0.51) | 65.16(0.53) | 63.34(0.45) | 70.06(1.45) | 81.42(2.65) | 77.71(0.45) | 81.32(1.25) |
| | + GPF | 66.83 (0.86) | 79.09 (0.25) | 66.10 (0.53) | 66.17 (0.81) | 73.56 (3.94) | 80.43 (0.53) | 76.49 (0.18) | 83.60 (1.00) |
| | + GPF-plus | 67.17 (0.36) | 79.13(0.70) | 66.35(0.37) | 65.62 (0.74) | 75.12(2.45) | 81.33 (1.52) | 77.73(1.14) | 83.67(1.08) |
| | **+ Ours** | **69.24(0.84)** | **80.18(0.25)** | **67.10 (0.38)** | **66.40(0.36)** | **80.02 (1.3)** | **82.84(0.58)** | **78.77(0.60)** | **83.88(0.45)** |
| AttrMasking | +FT | 66.33 (0.55) | 78.28 (0.05) | 65.34 (0.42) | 66.77 (1.02) | 74.46 (2.82) | 81.78 (1.95) | 77.90 (0.18) | 80.94 (1.17) |
| | +GPF | 68.09 (0.38) | 79.04 (0.90) | 66.32 (0.42) | 69.13 (1.16) | 75.06 (1.02) | 82.17 (0.65) | 78.86 (1.42) | 84.33 (0.54) |
| | +GPF-plus | 67.71 (0.64) | 78.87 (0.31) | 66.58 (0.13) | 68.65 (0.72) | 76.17 (2.98) | 81.12 (1.32) | 78.13 (1.12) | **85.76 (0.36)** |
| | **+ Ours** | **69.56 (0.82)** | **80.77 (0.43)** | **67.91 (0.38)** | **69.15(0.40)** | **85.18 (0.12)** | **85.01(0.61)** | **79.48(0.64)** | 83.92 (1.46) |
| ContextPred | +FT | 69.65 (0.87) | 78.29 (0.44) | 66.39 (0.57) | 64.45 (0.60) | 73.71 (1.57) | 82.36 (1.22) | 79.20 (0.51) | 84.66 (0.84) |
| | +GPF | 68.48 (0.88) | 79.99 (0.24) | 67.92 (0.35) | 66.18 (0.53) | 74.51 (2.72) | 84.34 (0.25) | 78.62 (1.46) | 85.32 (0.41) |
| | +GPF-plus | 69.15 (0.82) | 80.05 (0.46) | 67.58 (0.54) | 66.94 (0.95) | 75.25 (1.88) | 84.48 (0.78) | 78.40 (0.16) | 85.81 (0.43) |
| | **+ Ours** | **73.62 (0.11)** | **80.60 (0.27)** | **68.81 (0.6)** | **68.52 (0.3)** | **81.21 (0.86)** | 84.96 (1.2) | **81.76 (0.10)** | **86.57 (0.49)** |
| EdgePred | +FT | 66.56 (3.56) | 78.67 (0.35) | 66.29 (0.45) | 64.35 (0.78) | 69.07 (4.61) | 79.67 (1.70) | 77.44 (0.58) | 80.90 (0.92) |
| | GPF | 69.57 (0.21) | 79.74 (0.03) | 65.65 (0.30) | 67.20 (0.99) | 69.49(5.17) | 82.86 (0.23) | 77.00 (1.08) | 81.57 (1.08) |
| | +GPF-plus | 69.06(0.68) | 80.04 (0.06) | 65.94 (0.31) | 67.51(0.59) | 68.80 (2.58) | 83.13(0.42) | 77.00 (0.82) | 81.75(2.09) |
| | **+ Ours** | **72.11(0.50)** | **81.26(0.80)** | **67.05(0.60)** | **67.25(0.30)** | **77.68(1.10)** | **86.30(1.20)** | **79.38(0.01)** | **81.79(0.11)** |

an ROC-AUC of 73.62 under the ContextPred pretraining strategy, reflecting an 8.98% improvement over FineTune. Similarly, on ClinTox, it attains 85.18 under AttrMasking, surpassing GPF-plus by 11.83%. On MUV, despite FT's strong baseline performance, our method reaches 86.30, yielding a 6.00% improvement. These results highlight its superior performance and generalizability, consistently surpassing alternative approaches across diverse datasets and pre-training strategies. Its strong performance on complex or low-quality datasets further underscores its robustness and adaptability in graph-related tasks.

We also compared our model with other SOTA models for molecular property prediction. As shown in Table 2, our method outperforms SOTA models in 5 out of 8 tasks while using fewer parameters, demonstrating its efficiency and effectiveness.

Further analysis reveals that our approach offers distinct advantages over prompt-tuning models in molecular property prediction. Specifically, the MIPT framework effectively captures multi-level structural features in molecular data. Its novel prompt-learning paradigm facilitates knowledge transfer from pre-training to downstream tasks, mitigating distributional and semantic discrepancies between pre-training and target scenarios.

### 5.3. Additional Experiments

**Transfer Performance Analysis (RQ2).** We analyzed the training process of the molecular dataset and the GNN model using different tuning methods. Figure 2 (a) shows the testing curves during the tuning phase. From the curve, it is evident that our method exhibits greater fluctuations on

the test dataset compared to other models. This suggests that the model is more sensitive to the data during training, allowing it to flexibly adjust and capture complex patterns in the dataset. The increased fluctuations indicate a more exploratory optimization process, which facilitates finding a better global solution. Additionally, this behavior demonstrates strong adaptability and robustness to complex data distributions and noise. More training curves are provided in Appendix D.

**Ablation Study (RQ3).** To assess the effect of each component in our method, we conduct an ablation study evaluating LoRA, MGIP, and the loss functions ($\mathcal{L}_{NPP}$ and $\mathcal{L}_{cont}$). The results in Table 3 highlight the significance of MGIP over LoRA when used independently, achieving an average improvement of 6.9% on SIDER and 1.6% on BACE , demonstrating its superior ability to capture task-specific features. Integrating $\mathcal{L}_{cont}$ with LoRA further enhances performance, yielding a 6.5% gain on SIDER, indicating that contrastive learning effectively aligns global and local representations. Similarly, incorporating $\mathcal{L}_{NPP}$ into MGIP boosts ClinTox performance by 1.2%, underscoring the importance of node-to-graph alignment in refining global representations.

When combining all components, the complete model achieves the best results across all datasets, with improvements of 2.9% on SIDER, 1.3% on ClinTox, and 0.9% on BACE compared to the best individual configurations. These results confirm the complementary nature of the components and the effectiveness of the proposed approach in addressing multi-scale representation challenges in graph learning. Additional ablation results on various datasets are presented in Appendix D Table 6.

*Table 2.* Performance comparison with the SOTA models on molecular property prediction datasets(ROC-AUC %, higher is better ↑)
.

| Pretraining | Datasets↑ | BBBP↑ | BACE↑ | ClinTox↑ | Tox21↑ | ToxCast↑ | SIDER↑ | MUV↑ | HIV↑ |
|---|---|---|---|---|---|---|---|---|---|
| | # Molecules | 2039 | 1513 | 1477 | 7831 | 8576 | 1427 | 93087 | 41127 |
| | # Tasks | 1 | 1 | 2 | 12 | 617 | 27 | 17 | 1 |
| ✗ | D-MPNN | 71.0 (0.3) | 80.9 (0.6) | 90.6 (0.6) | 75.9 (0.7) | 65.5 (0.3) | 57.0 (0.7) | 77.1 (0.5) | 78.6 (1.4) |
| | Attentive FP | 64.3 (1.8) | 78.4 (0.02) | 84.7 (0.3) | 76.1 (0.5) | 63.7 (0.2) | 60.6 (3.2) | 75.7 (1.4) | 76.6 (1.5) |
| | MGCN | 65.0 (0.5) | 73.4 (0.8) | 90.5 (0.4) | 74.1 (0.6) | - | 58.7 (1.9) | - | - |
| | N-Gram$_{RF}$ | 69.7 (0.6) | 77.9 (1.5) | 77.5 (4.0) | 74.3 (0.4) | - | 66.8 (0.7) | 77.2(0.1) | 76.9(0.7) |
| | N-Gram$_{XGB}$ | 69.1 (0.8) | 79.1 (1.3) | 87.5 (2.7) | 75.8 (0.9) | - | 65.5 (0.7) | 78.7(0.4) | 74.8(0.2) |
| | PretrainGNN | 68.7 (1.3) | 84.5 (0.7) | 72.6 (1.5) | 78.1 (0.6) | 65.7 (0.6) | 62.7 (0.8) | 79.9(0.7) | 81.3(2.1) |
| | GROVER$_{base}$ | 70.0 (0.1) | 82.6 (0.7) | 81.2 (3.0) | 74.3 (0.1) | 65.4 (0.4) | 64.8 (0.6) | 62.5(0.9) | 67.3(1.8) |
| | GROVER$_{large}$ | 69.5 (0.1) | 81.0 (1.4) | 76.2 (3.7) | 73.5 (0.1) | 65.3 (0.5) | 65.4 (0.1) | 68.2(1.1) | 67.3(1.8) |
| ✓ | 3D-Infomax | 69.1 (1.1) | 79.4 (1.9) | 59.4 (3.2) | 74.5 (0.7) | 64.4 (1.0) | 53.3 (3.4) | | |
| | GraphMVP | 72.4 (1.6) | 81.2 (0.9) | 79.1 (2.8) | 75.9 (0.5) | 63.1 (0.4) | 63.9 (1.2) | 77.0(1.2) | 77.7(0.6) |
| | MolCLR | 72.2 (2.1) | 82.4 (0.9) | 91.2 (3.5) | 75.0 (0.2) | - | 58.9 (1.4) | 78.1(0.5) | 79.6(1.9) |
| | Uni-Mol | 72.9 (0.6) | 85.7 (0.2) | 91.9 (1.8) | 79.6 (0.5) | 69.6 (0.1) | 65.9 (1.3) | 80.8(0.3) | **82.1(1.3)** |
| | GEM | 72.4 (0.4) | 85.6 (1.1) | 90.1 (1.3) | 78.1 (0.1) | 69.2 (0.4) | 67.2 (0.4) | 80.6(0.9) | 81.7(0.5) |
| | InstructMol | 73.3(0.8) | 85.9(1.3) | **92.5(2.1)** | 79.9(0.6) | **70.8(0.4)** | 67.4(0.9) | - | - |
| | **Ours** | **73.6(0.1)** | **86.6(0.5)** | 85.2(0.1) | **80.6 (0.3)** | 68.8(0.6) | **68.5 (0.3)** | **85.0 (1.2)** | 81.8 (0.1) |

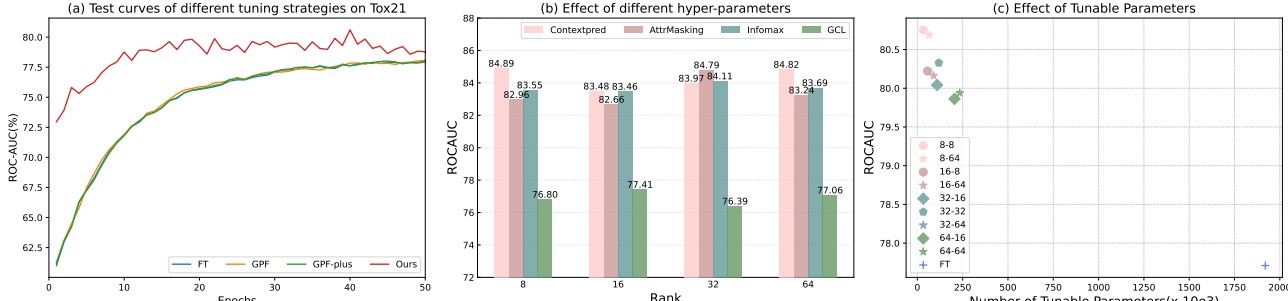

*Figure 2.* (a) Test curves for various tuning methods. (b) Impact of different hyperparameters, where the y-axis denotes the ROC-AUC score (%) and the x-axis represents the LoRA rank of node encoder hyperparameters. (c) Comparison of ROC-AUC (%) and trainable parameter count between MIPT with varying rank values and full fine-tuning on the Tox21 dataset.

*Table 3.* Ablation analysis of different configurations on BBBP, ClinTox, BACE, and SIDER datasets.

| Module | | Loss | | Datasets | | | |
|---|---|---|---|---|---|---|---|
| LoRA | MGIP | $\mathcal{L}_{NPP}$ | $\mathcal{L}_{cont}$ | bbbp | ClinTox | BACE | Sider |
| ✓ | – | – | ✓ | 68.72 | 67.76 | 86.41 | 60.75 |
| – | ✓ | – | ✓ | 72.45 | 80.19 | 85.22 | 67.17 |
| ✓ | ✓ | – | ✓ | 72.88 | 81.18 | 86.42 | 67.76 |
| ✓ | – | ✓ | ✓ | 69.43 | 70.99 | 85.51 | 61.26 |
| – | ✓ | ✓ | ✓ | 72.69 | 80.27 | 85.46 | 67.23 |
| ✓ | ✓ | ✓ | – | 73.18 | 80.70 | 85.51 | 66.17 |
| ✓ | ✓ | ✓ | ✓ | **73.62** | **81.21** | **86.52** | **68.52** |

**Hyperparameter Analysis (RQ4).** The test curves in Figure 2 (a) show that our method achieves higher ROC-AUC scores and remains stable across training epochs, outperforming FT, GPF, and GPF-plus, demonstrating the effectiveness of low-rank adaptation for generalization and efficiency. Figure 2 (b) compares the performance of four pre-trained models on the Tox21 dataset under different

LoRA ranks (8, 16, 32, 64), showing that lower ranks maintain high performance while reducing parameter overhead, highlighting the efficiency of low-rank adaptation. Figure 2 (c) illustrates the trade-off between trainable parameters and performance, confirming that our approach significantly reduces computational costs without sacrificing accuracy compared to full fine-tuning. These results demonstrate that low-rank prompts provide a scalable and efficient alternative to traditional fine-tuning. Additionally, analysis of label distributions across datasets (Appendix C.1) underscores our method's robustness to noisy data and adaptability to real-world scenarios.

## 6. Conclusion

In this work, we propose Multilevel Informed Prompt Tuning (MIPT), a novel framework for enhancing pre-trained molecular encoders in molecular property prediction tasks. MIPT employs a lightweight, multi-level prompt learning

network and a noise penalty mechanism to bridge the gap between pre-training and downstream tasks, while mitigating noise at both node and graph levels. Experiments on public and real-world datasets demonstrate MIPT's superior performance, highlighting its potential to advance drug discovery and materials science.

## Impact Statement

This paper presents work whose goal is to advance the field of Machine Learning. There are many potential societal consequences of our work, none of which we feel must be specifically highlighted here.

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

## A. Proof: Effectiveness of LoRA Updates in GNNs

Consider a GNN layer, the feature update at layer $k + 1$ for node $v$ is expressed as:

$$h_v^{(k+1)} = \sigma \left( \mathbf{W} \cdot h_v^{(k)} + \text{AGG} \left( \left\{ h_u^{(k)} : u \in \mathcal{N}(v) \right\} \right) \right), \tag{22}$$

where $h^{(k)} \in \mathbb{R}^{n \times d}$ represents the node embeddings at layer $k$, and $W \in \mathbb{R}^{d \times d}$ is a trainable weight matrix. The function $\sigma(\cdot)$ denotes a non-linearity such as ReLU.

Let $M \in \mathbb{R}^{n \times d}$ stack all aggregated messages, and let $\delta \in \mathbb{R}^{n \times d}$ be the gradient of the loss w.r.t. outputs. Then:

$$g = \nabla_W \mathcal{L} = \delta^\top M \tag{23}$$

Since $M$ results from summing over neighbors in sparse graphs, it is typically low-rank or approximately so. Thus, $g$ is approximately low-rank. From the **Eckart-Young-Mirsky theorem**, the best rank-$r$ approximation of $g$ under Frobenius norm is:

$$\Delta W^* = \sum_{i=1}^{r} \sigma_i u_i v_i^\top, \tag{24}$$

where $g = U\sigma V^\top$ is the SVD of $gradient$, and $\sigma_i$ are the top $r$ singular values. This can be written as $\Delta W = AB$ with:

$$A = U_r \sigma_r^{1/2} \in \mathbb{R}^{d \times r}, \quad B = \sigma_r^{1/2} V_r^\top \in \mathbb{R}^{r \times d}, \tag{25}$$

which is precisely the LoRA parameterization. Following PAC-Bayes or norm-based generalization analysis:

$$\mathbb{E}_{\mathcal{D}}[\mathcal{L}_{\text{test}}] \leq \mathbb{E}_{\mathcal{D}}[\mathcal{L}_{\text{train}}] + O \left( \frac{\|\Delta W\|_F^2}{n} \right), \tag{26}$$

where n is the number of training samples and $\| \cdot \|_F$ denotes the Frobenius norm. Since $\|\Delta W\|_F^2 = \|AB\|_F^2 \ll \|G\|_F^2$, the low-rank constraint imposed by LoRA acts as an implicit regularizer, effectively reducing model complexity and improving generalization performance.

Then, the updated node feature equation becomes:

$$h_v^{(k+1)} = \sigma \left( (\mathbf{W}_0 + \mathbf{AB}) \cdot h_v^{(k)} + \text{AGG} \left( \left\{ h_u^{(k)} : u \in \mathcal{N}(v) \right\} \right) \right). \tag{27}$$

The key point is that the neighborhood $\mathcal{N}(v)$ is solely determined by the static graph $G = (V, E)$, which can be described via the adjacency matrix $\mathbf{A_{adj}}$:

$$A_{adj}(u, v) = \begin{cases} 1, & \text{if } u \in \mathcal{N}(v) \\ 0, & \text{otherwise} \end{cases} \tag{28}$$

and does not depend on the model parameters. $LoRA$ modifies only the linear transformation term and does not alter the adjacency matrix or neighborhood relations. Therefore, $\mathcal{N}(v)$ are unaffected by the addition of $\mathbf{AB}$.

## B. Proof: Robustness of NPP

### B.1. Derivation of Posterior Probability $p(t \mid \mathcal{L}_{\text{cls}}^i)$

Let $\mathcal{L}_{\text{cls}} = \{\mathcal{L}_{\text{cls}}^1, \mathcal{L}_{\text{cls}}^2, \dots, \mathcal{L}_{\text{cls}}^N\}$ represent the observed classification loss values, and assume that each sample $\mathcal{L}_{\text{cls}}^i$ follows a Gaussian Mixture Model (GMM) distribution. For a GMM with $T$ components, the parameters include:

- $\pi_t$: The mixture coefficient for the $t$-th component.

- $\mu_t$: The mean of the $t$-th Gaussian component.

- $\sigma_t^2$: The variance of the $t$-th Gaussian component.

The posterior probability $p(t \mid \mathcal{L}_{\text{cls}}^i)$ can be computed using Bayes' rule:

$$p(t \mid \mathcal{L}_{\text{cls}}^i) = \frac{p(\mathcal{L}_{\text{cls}}^i \mid t) \cdot \pi_t}{\sum_{j=1}^{T} p(\mathcal{L}_{\text{cls}}^j \mid j) \cdot \pi_j}. \tag{29}$$

For the $k$-th Gaussian component, the likelihood term is given by:

$$p(\mathcal{L}_{\text{cls}}^i \mid t) = \frac{1}{\sqrt{2\pi\sigma_t^2}} \exp\left(-\frac{(\mathcal{L}_{\text{cls}}^i - \mu_t)^2}{2\sigma_t^2}\right). \tag{30}$$

Combining the likelihood and the prior, the posterior probability becomes:

$$p(t \mid \mathcal{L}_{\text{cls}}^i) = \frac{\pi_t \cdot \frac{1}{\sqrt{2\pi\sigma_t^2}} \exp\left(-\frac{(\mathcal{L}_{\text{cls}}^i - \mu_t)^2}{2\sigma_t^2}\right)}{\sum_{j=1}^{T} \pi_j \cdot \frac{1}{\sqrt{2\pi\sigma_j^2}} \exp\left(-\frac{(\mathcal{L}_{\text{cls}}^i - \mu_j)^2}{2\sigma_j^2}\right)}. \tag{31}$$

## B.2. EM Updates

E-STEP: COMPUTE POSTERIOR PROBABILITIES

It aims to estimate the posterior distributions of the loss, i.e. $P(\mathcal{L}_{\text{cls}}^i \mid \mu, \sigma)$, by using the current estimated parameters $\theta_{old} : \{\mu_{(old)}, \sigma^2\}$. Hence, we compute the posterior probabilities $p(t \mid \mathcal{L}_{\text{cls}}^i)$ using the current estimates of $\pi_t^{(n)}$, $\mu_t^{(n)}$, and $\sigma_t^{(n)}$:

$$p^{(n)}(t \mid \mathcal{L}_{\text{cls}}^i) = \frac{\pi_t^{(n)} \cdot p(\mathcal{L}_{\text{cls}}^i \mid \mu_t^{(n)}, \sigma_t^{(n)})}{\sum_{j=1}^{T} \pi_j^{(n)} \cdot p(\mathcal{L}_{\text{cls}}^i \mid \mu_j^{(n)}, \sigma_j^{(n)})}. \tag{32}$$

M-STEP: PARAMETER UPDATES

Using the posterior probabilities $p^{(n)}(t \mid \mathcal{L}_{\text{cls}}^i)$, we update the parameters as follows:

- **Mixture Coefficient:**

$$\pi_t^{(n+1)} = \frac{\sum_{i=1}^{N} p^{(n)}(t \mid \mathcal{L}_{\text{cls}}^i)}{N}. \tag{33}$$

- **Mean:**

$$\mu_t^{(n+1)} = \frac{\sum_{i=1}^{N} p^{(n)}(t \mid \mathcal{L}_{\text{cls}}^i) \cdot \mathcal{L}_{\text{cls}}^i}{\sum_{i=1}^{N} p^{(n)}(t \mid \mathcal{L}_{\text{cls}}^i)}. \tag{34}$$

- **Variance:**

$$\sigma_t^{2(n+1)} = \frac{\sum_{i=1}^{N} p^{(n)}(t \mid \mathcal{L}_{\text{cls}}^i) \cdot (\mathcal{L}_{\text{cls}}^i - \mu_t^{(n+1)})^2}{\sum_{i=1}^{N} p^{(n)}(t \mid \mathcal{L}_{\text{cls}}^i)}. \tag{35}$$

## B.3. Proof of NPP Robustness

The Noise Prompt Penalty (NPP) framework leverages the posterior probability $p(t \mid \mathcal{L}_{\text{cls}}^i)$ to compute the weight $w_i$ for each sample:

$$w_i = p(t \mid \mathcal{L}_{\text{cls}}^i). \tag{36}$$

By thresholding the weights $w_i$ based on the mean value, we define:

$$\hat{w}_i = \begin{cases} 1, & \text{if } w_i > \text{mean}(w_i), \\ 0, & \text{otherwise.} \end{cases} \tag{37}$$

This filtering ensures that low-confidence (likely noisy) samples are excluded from the loss computation. The final NPP loss

---

**Algorithm 1** Noise Penalty Sample Training Procedure

---

**Input:** Downstream task model $g_\theta$, labeled data $\mathcal{D}$, classification loss $\mathcal{L}_{cls}$, noise prompts penalty loss $\mathcal{L}_{NPP}$, learning rate $\eta$, noise weight $\lambda$, prompt-related parameters $p_g$, LoRA parameters $\delta$. Initialize model parameters $\theta$, GMM parameters $\pi_t, \mu_t, \sigma_t$.

**Output:** Optimized model parameters $\theta$

1: for $t = 1, 2$
2: **for** *epoch=1:max epoch* **do**
3:    **for** each sample $(G_i, y_i) \in \mathcal{D}$ **do**
4:       Compute prediction: $\hat{y}_i \leftarrow g_\theta(G_i)$
5:       Compute classification loss using Eq. 14;
6:    **end for**
7:    GMM Modeling: $w_i \leftarrow \text{GMM}(\mathcal{L}_{cls}^i)$
8:    **while** not converged or max iterations reached **do**
9:       **Update $w_i$ using EM algorithm**
10:    **end while**
11:    Update confidence $\hat{w}_i$ using Eq. 17
12:    Compute noise penalty loss using Eq. 13;
13:    **for** iteration from 1 to $T$ **do**
14:       $\mathcal{L}_{total} = \mathcal{L}_{cont} + \mathcal{L}_{Npp}$;
      optimize the model parameters $\theta$ by $\mathcal{L}_{total}$
15:    **end for**
16: **end for**
17: **Return** $\theta, p_g, \delta$

---

is computed as:

$$\mathcal{L}_{\text{NPP}} = -\frac{1}{N} \sum_{i=1}^{N} \hat{w}_i \log P\big(y^i \mid g(h_{G,i})\big). \tag{38}$$

This framework achieves robustness by:

- Utilizing the EM algorithm to iteratively refine the estimation of GMM parameters $(\pi_t, \mu_t, \sigma_t^2)$, leading to improved separation of noise and trustworthy samples.

- Filtering out low-confidence samples ($\hat{w}_i = 0$), ensuring that only high-confidence samples contribute to the model optimization.

The robustness of NPP is thus guaranteed by the iterative EM updates and selective loss computation, effectively mitigating the impact of noisy samples. Pseudocode is presented in Algorithm 1.

## C. More Experiment Settings.

### C.1. Details of Datasets.

**Pre-training datasets.** The dataset includes 2 million unlabeled molecules drawn from the ZINC15 database, used for node-level self-supervised pre-training. For graph-level multi-task supervised pre-training, the preprocessed ChEMBL dataset is utilized, consisting of 456K molecules spanning 1310 distinct biochemical assays.

**Downstream Task.** To evaluate model performance, we utilized eight binary graph classification datasets from MoleculeNet (Wu et al., 2018), described as follows:

- **BBBP** (Martins et al., 2012): Evaluates blood-brain barrier penetration based on membrane permeability.

- **Tox21** (tox): Provides toxicity data for 12 biological targets, including nuclear receptors and stress response pathways.

- **ToxCast** (Richard et al., 2016): Contains toxicological measurements derived from over 600 in vitro high-throughput screening assays.

- **SIDER**: A comprehensive database of marketed drugs and their associated adverse drug reactions (ADR), categorized into 27 system organ classes.

- **ClinTox**: Includes qualitative data distinguishing FDA-approved drugs from those that failed clinical trials due to toxicity.

- **MUV**: A subset of PubChem BioAssay data, specifically curated using a refined nearest-neighbor analysis for validating virtual screening techniques.

- **HIV**: Contains experimental data assessing the inhibitory efficacy of compounds against HIV replication.

- **BACE** (Subramanian et al., 2016): Provides qualitative binding data for inhibitors targeting human $\beta$-secretase 1.

*Table 4.* Dataset statistics.

| Dataset | BBBP | Tox21 | ToxCast | SIDER | Clintox | MUV | HIV | BACE |
|---|---|---|---|---|---|---|---|---|
| # Molecule | 2039 | 7831 | 8575 | 1427 | 1478 | 93127 | 41127 | 1513 |
| # Property | 1 | 12 | 617 | 27 | 2 | 17 | 1 | 1 |
| % Positive Label | 76.44 | 6.24 | 12.60 | 56.76 | 50.61 | 0.31 | 96.49 | 45.67 |
| % Negative Label | 23.56 | 76.71 | 72.43 | 43.24 | 49.39 | 15.76 | 1.18 | 54.33 |
| % Unknown Label | 0 | 17.05 | 14.97 | 0 | 0 | 84.21 | 2.33 | 0 |

## C.2. Details of pre-training strategies.

We utilize five widely recognized strategies for pre-training GNN models, detailed as follows:

- **Deep Graph Infomax**: Initially proposed by Velickovic et al. (Veličković et al., 2018), this method acquires expressive representations for graphs or nodes by maximizing mutual information between graph-level and substructure-level representations across various granularities.

- **Edge Prediction**: A common graph reconstruction task employed by many models, such as GAE (Kipf & Welling, 2016), where the goal is to predict the existence of edges between pairs of nodes.

- **Attribute Masking**: Introduced by Hu et al. (Hu et al., 2019), this technique involves masking node/edge attributes and then using GNNs to predict these attributes based on the neighboring structure.

- **Context Prediction**: Also proposed by Hu et al. (Hu et al., 2019), this approach uses subgraphs to forecast their surrounding graph structures, aiming to map nodes in similar structural contexts to nearby embeddings.

In our model pre-training phase, we adhere to the methodologies described by Hu et al. (2019) for the tasks of Infomax, EdgePred, AttrMasking, and ContextPred. Subsequently, we engage in supervised graph-level property prediction to further boost the efficacy of our pre-trained models.

**Training Details.** We perform five rounds of experiments with different random seeds for each experimental setting and report the average results. The projection head $\theta$ is selected from a range of [1, 2, 3]-layer MLPs with equal widths. The hyper-parameter $k$ of GPF-plus is chosen from the range [5,10,20]. Further details on the hyper-parameter settings can be found in the appendix.

**Implementation Details.** All experiments were conducted on a high-performance computing server equipped with an NVIDIA 3090 GPU (24 GB memory). The implementation was based on Python 3.9, PyTorch 1.12, and the torc_geometric library. For the GNN architecture, we utilized Graph Isomorphism Network (GIN), configured with a hidden dimension of 300, 3 graph convolutional layers, ReLU activation, and batch normalization. The optimizer was Adam with a learning rate of 0.001, dropout rate is 0.5, the mask probability is 0.2. Pretraining tasks included node-level attribute prediction and

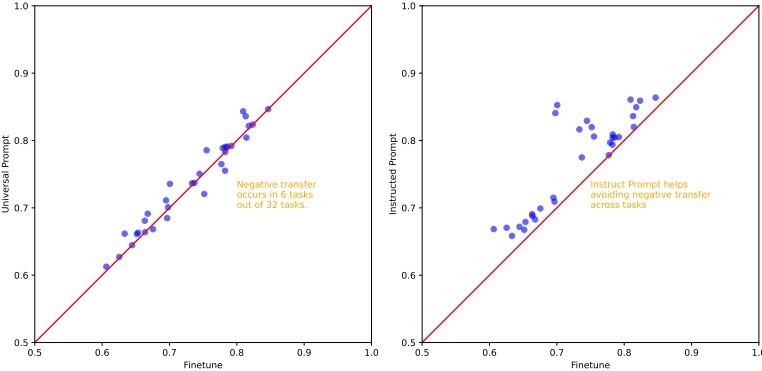

*Figure 3.* Test ROC-AUC of molecule property prediction using different tuning strategies with different pre-training strategies. Each point represents a particular individual downstream task from (Fang et al., 2024) (left) and our method (right).

graph-wise classification, with data augmentation strategies such as node masking, edge sampling, and feature perturbation applied during training. The experiments were conducted on MoleculeNet datasets, running for 100 epochs with a batch size of 32. Performance was evaluated using ROC-AUC, and each experiment was repeated five times to ensure reproducibility, with the mean and standard deviation reported.

## D. Additional Experiments

### D.1. More comparison experiments

Figure 3 compares the test ROC-AUC performance of molecular property prediction across different tuning strategies with various pre-training approaches. Each point represents the performance of an individual downstream task, with GPF shown on the left and our method on the right. The red diagonal line indicates parity, where points above the line represent improved performance compared to fine-tuning.

The left plot reveals that the method from GPF exhibits negative transfer in several tasks, as some points fall below the diagonal, indicating a degradation in performance due to ineffective knowledge transfer. In contrast, the right plot demonstrates that our proposed Instruct Prompt effectively mitigates negative transfer, resulting in a more consistent improvement across tasks. This highlights the robustness of our approach in enhancing molecular representation learning and improving generalization across diverse downstream tasks.

### D.2. More training cases.

We provide more training curves in Figure 4 as a supplement to Section 5.3. Our method achieves a considerable performance improvement over other approaches during training. We note that on some smaller datasets, such as Clintox and BBBP, the training curves exhibit greater fluctuations. This behavior is hypothesized to stem from the increased number of parameters in our model coupled with the constrained data availability. Crucially, despite these fluctuations, overfitting is not observed, suggesting that the model remains robust. In stark contrast, training on larger datasets yields much smoother performance curves, with fluctuations being significantly less prominent.

*Table 5.* The number of tunable parameters for different tuning strategies. * is the frozen parameters.

|  | FT | GPF | GPF-plus | Ours |
|---|---|---|---|---|
| Size of GNN(Encoder) | 1.86M | 1.86M* | 1.86M* | 1.86M* |
| Size of Prompt | 0 | 0.3K | 3-12K | 0.115M |
| Size of Graph Linear Layer | 0.6K-0.18M | 0.6K-0.18M | 0.6K-0.18M | 0.6K-0.18M |
| Size of Total Tunable Part | 1.92M-2.14M | 0.9K-0.21M | 3.6K-0.192M | 0.175M-0.295M |
| # Params(%) | 100 | 0.046-0.98 | 0.187-8.97 | 9.1-13.78 |

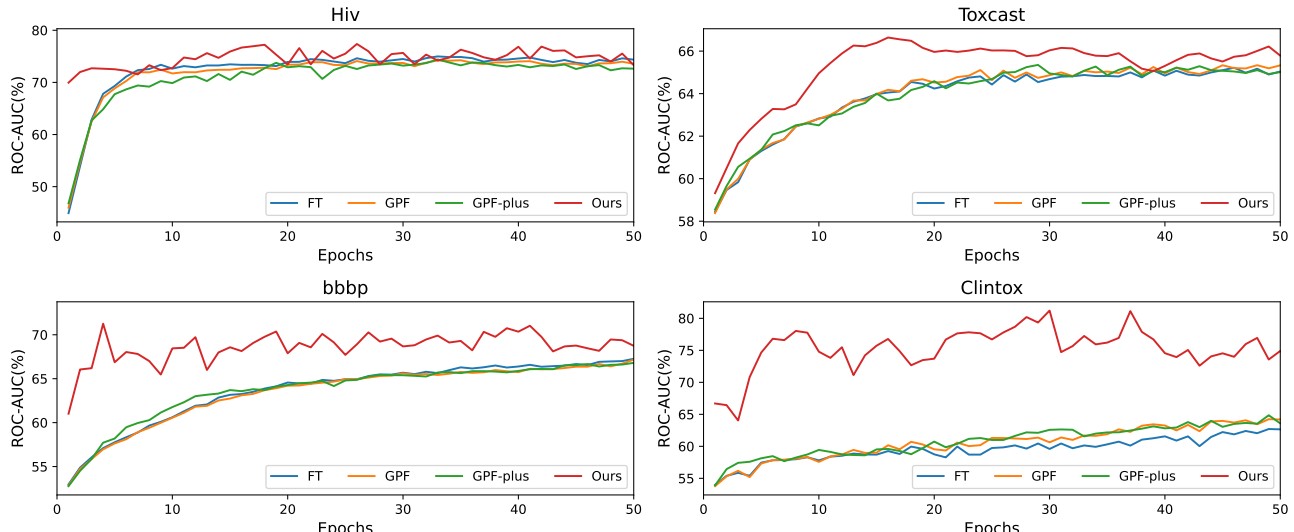

*Figure 4.* Test curves of different tuning strategies on pretrained GINs. Solid and dashed lines indicate training and validation curves, respectively.

*Table 6.* Ablation analysis of different configurations on Tox21, Toxcast, HIV, MUV, BBBP, Clintox, BACE and SIDER datasets.

| MODULE | | LOSS | | DATASETS | | | | | | | |
|---|---|---|---|---|---|---|---|---|---|---|---|
| LORA | MGIP | $\mathcal{L}_{NPP}$ | $\mathcal{L}_{\text{CONT}}$ | TOX21 | TOXCAST | HIV | MUV | BBBP | CLINTOX | BACE | SIDER |
| ✓ | – | – | ✓ | 75.31 | 64.34 | 75.95 | 82.44 | 68.72 | 67.76 | 86.41 | 60.75 |
| – | ✓ | – | ✓ | 80.57 | 68.78 | 79.97 | 84.47 | 72.45 | 80.19 | 85.22 | 67.17 |
| ✓ | ✓ | – | ✓ | 80.19 | 67.70 | 78.50 | 83.07 | 72.88 | 81.18 | 86.42 | 67.76 |
| ✓ | – | ✓ | ✓ | 76.40 | 68.55 | 75.03 | 79.48 | 69.43 | 70.99 | 85.51 | 61.26 |
| – | ✓ | ✓ | ✓ | 80.29 | 68.27 | 80.42 | 84.71 | 72.69 | 80.27 | 85.46 | 67.23 |
| ✓ | ✓ | ✓ | – | 80.45 | 68.52 | 79.84 | 82.45 | 73.18 | 80.70 | 85.51 | 66.17 |
| ✓ | ✓ | ✓ | ✓ | **80.60** | **68.81** | **81.76** | **84.96** | **73.62** | **81.21** | **86.52** | **68.52** |

### D.3. More ablation study

Here, Table 6 shows the comprehensive ablation results on eight datasets. A comparison of the parameters of the different high-efficiency fine-tuning methods is shown in Table 5. From the experimental results, we can conclude that our method can achieve the optimal performance with fewer parameters, showing superiority.

## E. Limitations and Future Work

As discussed in Section 5, while our method significantly outperforms state-of-the-art baselines, it has certain limitations in few-shot learning. This is primarily due to the constraints of GMM modeling, which requires at least two samples per class for effective learning, limiting its applicability in few-shot scenarios. Additionally, GMM introduces some computational overhead, though its resource consumption remains modest. Furthermore, our method exhibits higher standard deviations across multiple runs with different ranks, likely due to model uncertainty in handling diverse graph structures. Addressing the integration of node-level and graph-level features remains an open challenge for future research.

