# OpenReview forum: "MIPT: Multilevel Informed Prompt Tuning for Robust Molecular Property Prediction"
_ICML.cc/2025/Conference — ICML 2025 poster_

### Official Review · Reviewer_dDZn · 2025-03-05

**Overall Recommendation:** 3

**Summary:**

The paper introduces Multilevel Informed Prompt Tuning (MIPT), a framework that enhances pretrained Graph Neural Networks for molecular property prediction. MIPT significantly outperforms existing methods, achieving higher ROC-AUC scores while reducing the number of trainable parameters. Key contributions include a multilevel prompt learning module for capturing task-specific knowledge, a noise penalty mechanism to mitigate irrelevant information, and low-rank adaptation for efficient tuning.

**Claims And Evidence:**

Yes.

**Essential References Not Discussed:**

No.

**Experimental Designs Or Analyses:**

1. LoRA has proven its effectiveness through ablation experiments, but why weren't other feature extraction methods used? Was there a comparison with other methods to make the final decision?
2. The title of Table 3 is inconsistent with its content. The title mentions a comparison on Tox21, but the final results are based on BBBP. Additionally, why wasn't the comparison conducted across all datasets?
3. Both the abstract and the method section mention the random noise mask, so why wasn't this discussed in the ablation study?

**Methods And Evaluation Criteria:**

Yes.

**Other Comments Or Suggestions:**

No.

**Other Strengths And Weaknesses:**

1. The H_v(k) in the line above Eq. 2 is incorrect.

**Questions For Authors:**

No.

**Relation To Broader Scientific Literature:**

1. Advances in GNNs
MIPT builds on recent advancements in GNN architectures, which have been shown to effectively capture complex relationships in molecular data.
2. Prompt Tuning Techniques
MIPT extends this prompt tuning to LoRA, contributing to the body of work that explores how prompt-based methods can enhance performance across various domains.

**Theoretical Claims:**

Yes.Mainly experiments.

---

> ### Author Rebuttal · Authors · 2025-04-01
>
> Dear reviewer dDZn:
>
> Thank you for your valuable and constructive comments. We have revised the paper according to your suggestions.
>
> ## Experimental Designs Or Analyses
>
> **1. Why choose LoRA**: To learn the features of both node-level and graph-level, we used multi-level fine-tuning, but in order not to increase the training cost, we chose lightweight LoRA. Although there are alternative methods such as Adapter tuning [1] and BitFit [2], previous study [3] have shown that LoRA achieves the best balance between efficiency and performance.
>
> [1] Parameter-Efficient Transfer Learning for NLP.
>
> [2] Bitfit: Simple parameter-efficient fine-tuning for transformer-based masked language-models.
>
> [3] LoRA: Low-Rank Adaptation of Large Language Models.
>
> **2. Not all datasets were evaluated:** We apologized for the mistake in title. Due to space, we did not include all the data, which we will add to the appendix. Here we supplement the performance results of other datasets.
>
> Table. Ablation analysis of different configurations on Tox21, Toxcast, HIV, and MUV datasets based on RUC-AOC(%).
> |   Module |  Loss  | Tox21 | Toxcast | HIV | MUV |
> |-------|-------|-------|-------|-------|-------|
> | LoRA | $L_{cont}$ | 75.31 | 64.34 | 75.95 | 82.44 |
> | MGIP | $L_{cont}$ | 80.57 | 68.78 | 79.97 | 84.47 |
> | LoRA+MGIP| $L_{cont}$ | 80.19 | 67.70 | 78.50 | 83.07 |
> | LoRA | $L_{NPP}$+$L_{cont}$ | 76.40 | 68.55 | 75.03 | 79.48 |
> | MGIP | $L_{NPP}$+$L_{cont}$ | 80.29 | 68.27 | 80.42 | 84.71|
> | LoRA+MGIP | $L_{NPP}$ | 80.45 | 68.52 |  79.84 | 82.45 |
> | LoRA+MGIP | $L_{NPP}$+$L_{cont}$ | 80.60 | 68.81 | 81.76 | 84.96 |
>
>
> **3. The ablation experiment does not include the random noise mask.** Two parts of our approach mention noise. The random node mask discussed in the method is used to augment the original features, which are included in the GIP and the ablation experiment (MGIP). The noise penalty mechanism, aimed at reducing the uncertainty of the method and increasing confidence, has been applied in the ablation analysis (NPP).
>
> We appreciate the reviewer for spending time reviewing our paper and offering valuable suggestions. If you have any further questions, please tell us and we are willing to address your concerns.

---

### Official Review · Reviewer_Cxne · 2025-03-07

**Overall Recommendation:** 4

**Summary:**

This manuscript introduces a novel Multilevel Informed Prompt Tuning (MIPT) framework designed to enhance pre-trained molecular encoders for molecular property prediction tasks. The key contributions include a multi-level prompt learning network and a noise penalty mechanism. The proposed prompt learning network effectively mitigates the gap between pre-training and downstream tasks, while the noise penalty mechanism addresses potential mismatches between pre-trained representations and task-specific representations. Extensive experiments on both public and real-world datasets demonstrate that MIPT achieves significant improvements in molecule-related tasks.

**Claims And Evidence:**

Yes, this manuscript provides extensive experimental results, including comparisons with baseline models, ablation studies, and hyperparameter experiments.

**Essential References Not Discussed:**

[1]  Pin-Tuning: Parameter-Efficient In-Context Tuning for Few-Shot Molecular Property Prediction.
[2] MMGNN:AMolecularMerged Graph Neural Network for Explainable Solvation Free Energy Prediction

**Experimental Designs Or Analyses:**

In Table 2, the performance improvement on some datasets is not significant. Additional analysis should be provided.

**Methods And Evaluation Criteria:**

Yes

**Other Comments Or Suggestions:**

The manuscript requires more careful proofreading to avoid some nitpicks. For example:
1. In line 24, "MIT" should be corrected to "MIPT".
2. In line 216, there is a missing space after "conditions".
3. In line 341, the last numerical value should not be in boldface.

**Other Strengths And Weaknesses:**

Strengths

1) The manuscript presents a clear and well-structured motivation, precisely identifying the key challenges of pre-trained GNNs in molecular property prediction. It highlights the limitations of existing methods and naturally introduces its core solution。

2) The manuscript proposes multi-level prompt learning to extract task-specific information at both the node and graph levels, enabling the model to better adapt to downstream tasks and effectively bridging the gap between pre-training and real-world applications.

3) The manuscript employs a Gaussian Mixture Model (GMM) to model the confidence distribution of samples, introducing a noise penalty mechanism to suppress irrelevant noise, allowing the model to focus on truly meaningful features and enhancing overall performance.

4) The proposed method achieves significant performance improvements in molecular property prediction tasks across multiple pre-trained models.

Weaknesses

1) LoRA, as a parameter-efficient fine-tuning method, has been widely adopted in the LLM domain. The manuscript should more clearly clarify LoRA's unique advantages in molecular property prediction tasks and how it has been specifically optimized for molecular graph data.

2) What is this work different from Pin-tuning [1]?

3) What is the role of \(\epsilon^{(k)}\) in Eq. (5)?

4) In Table 2, the performance improvement on some datasets is not significant. Additional analysis should be provided.

5) Algorithm 1 needs careful proofreading

6) The manuscript does not provide the code.

[1] Pin-Tuning: Parameter-Efficient In-Context Tuning for Few-Shot Molecular Property Prediction.

**Questions For Authors:**

Please refer to Other Strengths And Weaknesses.

**Relation To Broader Scientific Literature:**

The manuscript presents a clear and well-structured motivation, precisely identifying the key challenges of pre-trained GNNs in molecular property prediction.

**Theoretical Claims:**

The manuscript includes a theoretical proof related to the Noise Prompt Penalty (NPP) to demonstrate its robustness in mitigating the impact of noisy samples.

---

> ### Author Rebuttal · Authors · 2025-03-28
>
> Dear reviewer Cxne:
>
> Thank you for your valuable and constructive comments. We have revised the paper according to your suggestions.
>
> **W1: LoRA, as a parameter-efficient fine-tuning method, has been widely adopted in the LLM domain. The manuscript should more clearly clarify LoRA's unique advantages in molecular property prediction tasks and how it has been specifically optimized for molecular graph data.**
>
> Our work is the first to adopt LoRA for fine-tuning GNNs in molecular property prediction. In NLP, LoRA is typically employed to cut down training costs. However, our goal is to construct graph prompts that capture the relationships between graph structure and node features, thereby optimizing LoRA specifically for molecular graph data while maintaining parameter efficiency.
>
>
> **W2: Different from  Pin-tuning.** Pin-tuning is designed to be based on adaptor fine-tuning strategy for few-shot learning. Unlike Pin-Tuning, our method bridges pre-training and fine-tuning via multilevel graph prompts that align node/graph structures with task requirements. Additionally, we incorporate a noise penalty to mitigate mismatches between pretrained representations and downstream tasks.
>
> **W3: What is the role of $\epsilon^{(k)}$ in Eq. (5)?**
> In Eq. (5), $\epsilon^{(k)}$ is a learnable parameter that modulates the contribution of the node's own features relative to its neighbors during the update at the $k$-th layer. Essentially, it scales the self-feature term, allowing the network to adaptively balance the importance of a node's current representation with the aggregated information from its neighbors.
>
> **W4: In Table 2, the performance improvement on some datasets is not significant. Additional analysis should be provided.**
>
> We acknowledge that the performance improvement of our method is modest on some datasets. We are aware that public molecular datasets are limited, and different SOTA methods tackle these challenges in various ways. For instance, Uni-Mol incorporates 3D information, while InstructMol uses pseudo-labels to enhance model confidence. In contrast, our approach mainly aims to bridge the gap between pre-training and fine-tuning. Although our method may not achieve SOTA performance on every molecular property, the enhancements it brings in transferability and robustness are still of great significance.
>
> **W5: Algorithm proofreading.**
> We have thoroughly proofread and revised Algorithm 1 to ensure its clarity and correctness.
>
> **W6: The manuscript does not provide the code.**
> We will make the code publicly available after the submission is published.
>
> **Minor issues.** For typos, we’ve corrected error and updated the description in the paper.
>
> We appreciate the reviewer for spending time reviewing our paper and offering valuable suggestions. If you have any further questions, please tell us and we are willing to address your concerns.

---

### Official Review · Reviewer_kVKq · 2025-03-09

**Overall Recommendation:** 4

**Summary:**

The paper introduces a novel framework called Multilevel Informed Prompt Tuning (MIPT) aimed at enhancing the performance of pretrained GNNs in molecular property prediction tasks. MIPT employs a lightweight multilevel prompt learning module to capture task-specific knowledge at both node and graph levels, while incorporating a noise penalty mechanism to mitigate the impact of irrelevant information. Experimental results demonstrate that MIPT outperforms baseline models across various molecular tasks, showcasing its effectiveness, scalability, and broad applicability, while also highlighting areas for future research, particularly in few-shot learning and stability across different graph structures.

**Claims And Evidence:**

Yes

**Essential References Not Discussed:**

No

**Experimental Designs Or Analyses:**

This paper presents a comprehensive evaluation of multiple benchmarks, comparing not only the SOTA model that has been pre-trained and fine-tuned but also the SOTA model specifically for molecular property prediction. The experimental design is well-conceived and effectively addresses the research objectives.

**Methods And Evaluation Criteria:**

When comparing the fine-tuning methods, the authors only compared the FT and GPF/GPF-plus strategies, and the results of other PEFT strategies should be added to illustrate the superiority of the methods. In addition, the authors should add more benchmarks on graph prompts to compare.

**Other Comments Or Suggestions:**

**Minor Weaknesses**
- The molecular graph provided in the upper and lower sections of Figure 1 is not identical.
- “MIT” in L430 should be “MIPT”.
- L696, Implementation Details are misaligned.

**Other Strengths And Weaknesses:**

**Strengths**
-This paper proposes MIPT, including multi-level prompt tuning and noise penalty mechanism, which provide a new solution for molecular knowledge transfer.
- This paper is well-motivated and easy-to-follow.
- The contributions are significant and somewhat new.

**Weaknesses**
- In this paper, only the benchmark of classification tasks is studied, and it is necessary to further increase the generality of other types of tasks to illustrate tasks, such as regression tasks.
- The SOTA performance in Table 2 is not available on all datasets.
- The author does not have open source code.

I am willing to consider raising the score based on your rebuttal to the following questions:
- The two LoRA module in Sec.4.1 and Sec.4.2 should provide more details on how they differ.
- Authors should compare their work with other benchmarks about graph prompt.
- The LoRA have already been used in many tasks, such as NLP tasks. The technical novelty of this paper is limited.

**Questions For Authors:**

- What is the difference between $L_{NPP}$ and $L_{cls}$ in Figure 1?
- The author emphasizes the multi-level graph prompt, essentially using the LoRA module twice, so the increase in performance could be due to an increase in the amount of training parameters? Please specify.

**Relation To Broader Scientific Literature:**

This paper introduces the innovative concept of molecular prompt tuning, which effectively bridges the gap between pre-training and fine-tuning without altering the molecular structure, while also enhancing interpretability.

**Theoretical Claims:**

I carefully examined the theory and its proof and found no major errors.

---

> ### Author Rebuttal · Authors · 2025-03-31
>
> Dear reviewer kVKq:
>
> Thank you for your valuable and constructive comments. We have revised the paper according to your suggestions.
> ## Weakness
>
> **W1: In this paper, only the benchmark of classification tasks is studied, and it is necessary to further increase the generality of other types of tasks to illustrate tasks, such as regression tasks.**
>
> In our paper, while the primary focus was on benchmark classification tasks, we recognize the importance of demonstrating the generality of our approach across other types of tasks, such as regression. We extended our experiments to regression tasks on Table I. The supplementary results indicate that our method maintains its superiority even in the regression setting.
>
> Table I. Comparison of the performance of different tuning strategies on regression tasks based on MAE.
>
> |  Tuning Strategy  | ESOL | Lipo |
> |-------|-------|-------|
> | FT | 1.1262 | 0.6942|
> | GPF | 1.1125 | 0.6832 |
> | GPF-plus | 1.1104 | 0.6789 |
> | Ours | 0.6788 | 0.6702 |
>
> **W2: The SOTA performance in Table 2 is not available on all datasets.** Thank you for pointing this out. Indeed, in molecular representation learning, the scope and design of different models can vary significantly, often incorporating specialized features or data. For example, UniMol leverages 3D information, and InstructMol adopts pseudo-labeling strategies to enhance confidence. While these SOTA methods focus on addressing certain limitations in molecular representation, our primary goal is to bridge the gap between pre-training and fine-tuning. We believe that even if not all molecular properties achieve SOTA performance, they still hold significant value.
>
> **W3: The author does not have open source code.** We will make the code publicly available after the submission is published.
>
> **W4: The two LoRA module in Sec.4.1 and Sec.4.2 should provide more details on how they differ.**
> We emphasis that the node-level LoRA is used for the node features of the adaptive learning molecule, and the LoRA on the graph-level features is a variant of LoRA, which is used as a prompt for adaptive learning of graph-level features.
>
> **W5: More benchmarks about Graph Prompt-based Methods.**
> We have conducted a comprehensive comparison with existing graph prompt benchmarks. As shown in the Table II, our method outperforms both GPPT and GraphPrompt on several key datasets, particularly on BBBP, Toxcast, SIDER, HIV, and BACE.
>
> Table II.  Comparison of benchmarks about graph prompt for the models pre-trained by Edge Prediction based on RUC-AOC(%).
>
> |    | BBBP | Tox21| Toxcast| SIDER| Clintox| MUV| HIV| BACE|
> |-------|-------|-------|-------|-------|-------|-------|-------|-------|
> | GPPT | 64.13 | 66.41 | 60.34 | 54.86 | 59.81 | 63.05 | 60.54 | 70.85 |
> | GraphPrompt | 69.29 | 68.09 | 60.54 | 58.71 | 55.37 | 62.35 | 59.31 | 67.70 |
> | Ours | 72.73 | 80.82 | 67.44 | 79.46 | 79.29 | 80.02 | 78.68 | 82.91 |
>
> **W6: The LoRA have already been used in many tasks, such as NLP tasks. The technical novelty of this paper is limited.**
>
> We agree that LoRA is an excellent and widely-used technique in many domains, including NLP. However, our paper does not claim novelty for the LoRA component itself. Instead, our main contribution lies in multilevel graph informed prompt and noise penalization mechanisms. This combination uniquely bridges the gap between pre-training and fine-tuning in molecular graph learning.
>
> ## Questions
>
> **Q1: What is the difference between $L_{NPP}$ and $L_{cls}$ in Figure 1?**
>
> $L_{cls}$ (Eq. 14) is the standard classification loss (binary cross-entropy loss) used to train the model on the downstream task. In contrast, $L_{NPP}$ (Eq. 18) adds a Noise Penalty Mechanism to $L_{cls}$.
>
> **Q2: The author emphasizes the multi-level graph prompt, essentially using the LoRA module twice, so the increase in performance could be due to an increase in the amount of training parameters?**
>
> Table III. The number of tunable parameters for different tuning strategies.
> | Tuning Strategy | Size of Total Tunable Part |
> |----------------|---------------------|
> | FT            | 1.92M-2.14M         |
> | GPF          | 0.9K-0.21M          |
> | GPF-plus      | 3.6K-0.192M         |
> | Ours          | 0.175M-0.295M       |
>
> While it is true that our method employs the LoRA module at multiple levels, the increase in performance cannot be solely attributed to a higher number of tunable parameters. We had provided the parametric analysis in tab. 5 in the appendix. Although our approach uses more parameters than GPF/GPF-plus, it is still far below FT, which utilizes 1.92M-2.14M parameters yet performs worse. This clearly indicates that our performance gains stem from the unique integration of the multilevel graph-informed prompt and the noise penalization mechanism, which effectively bridges the gap between pre-training and fine-tuning in molecular graph learning.

---

### Official Review · Reviewer_Swha · 2025-03-18

**Overall Recommendation:** 4

**Summary:**

This paper addresses the challenge of prompt tuning for pretrained models in molecular property prediction. It introduces a multi-level prompt learning module to enhance task adaptation and a noise penalty mechanism to improve robustness, adaptability, and efficiency. Extensive experimental evaluations demonstrate the effectiveness and superiority of the proposed approach across various molecular prediction tasks.

**Claims And Evidence:**

Yes, this paper is well-written and claims are well-supported

**Essential References Not Discussed:**

no

**Experimental Designs Or Analyses:**

The experimental setup is well-structured and includes multiple baselines

**Methods And Evaluation Criteria:**

Yes, the evaluation is comprehensive and the experimental setup is reasonable, while I'm curious about the effectiveness of the proposed method on other graph-related applications.

**Other Comments Or Suggestions:**

Please refers to the above parts and also the questions.

**Other Strengths And Weaknesses:**

Please refers to the above parts and also the questions.

**Questions For Authors:**

1. This method does not appear to be specifically designed for molecular property prediction. Is the prompt-tuning approach also effective for other tasks?

2. How does the fine-tuning time compare across different fine-tuning strategies?

3. How much data is required to fine-tune the model effectively?

**Relation To Broader Scientific Literature:**

prompt learning

**Theoretical Claims:**

Yes.

---

> ### Author Rebuttal · Authors · 2025-03-31
>
> Dear reviewer Swha:
>
> Thanks for your positive review of our paper and for your thoughtful comments.
>
> **For W1 model versatility.**
>
> - Thank you for your positive feedback on our approach. We appreciate your recognition of the potential our MIPT framework for other graph-level applications.
>
> Below is a comparison of the performance based on *RMSE* of different tuning strategies on regression tasks.
> |  Tuning Strategy  | ESOL | Lipo |
> |-------|-------|-------|
> | FT | 1.1262 | 0.6942|
> | GPF | 1.1125 | 0.6832 |
> | GPF-plus | 1.1104 | 0.6789 |
> | Ours | 0.6788 | 0.6702 |
>
> - In our initial research, we focus on molecular property prediction, where MIPT demonstrated exceptional performance. To explore its broader applicability, we extended our method to regression tasks according to your constructive comments. The experimental results indicate that our approach maintains superior performance in these tasks. We remain committed to further investigating and validating the framework's performance across a wider range of graph-level tasks.
>
> **For W2 model complexity.** Since fine-tuning time is often difficult to compare fairly due to differences in hardware and system environment, we analyze the training costs based on the number of tunable parameters.
>
> Table I. The number of tunable parameters for different tuning strategies. \* is the frozen parameters.
>
> | Tuning Strategy | Size of GNN (Encoder) | Size of Prompt | Size of Graph Linear Layer | Size of Total Tunable Part |
> |----------------|---------------------|---------------|----------------------|----------------------|
> | FT            | 1.86M               | 0             | 0.6K-0.18M          | 1.92M-2.14M         |
> | GPF           | 1.86M*               | 0.3K         | 0.6K-0.18M          | 0.9K-0.21M          |
> | GPF-plus      | 1.86M*               | 3-12K        | 0.6K-0.18M          | 3.6K-0.192M         |
> | Ours          | 1.86M*               | 0.115M       | 0.6K-0.18M          | 0.175M-0.295M       |
>
> * **FT** requires the highest number of tunable parameters (1.92M-2.14M), leading to the highest training cost.
> * **GPF** and **GPF-plus** significantly reduce the number of tunable parameters by freezing the GNN encoder. This makes them more parameter-efficient compared to FT.
> * **Our method** also freezes the GNN encoder but introduces a larger prompt size (0.115M). Despite this, the total number of tunable parameters (0.175M-0.295M) remains significantly lower than FT, suggesting a balance between efficiency and effectiveness.
>
> In summary, ​**GPF and GPF-plus are the most parameter-efficient**​, while ​**our approach achieves a middle ground**​, tuning more parameters than GPF but significantly fewer than FT, likely leading to a moderate training cost.
>
> **For Q3. the datasize of fine-tune:** In our study, we use the entire training set during the fine-tuning phase. This approach, which fine-tunes only a subset of model parameters rather than the entire network (see our discussion in ​**Q2**​), is a common practice in this field. Prior works [1,2,3,4] also employ the full training dataset for model fine-tuning, ensuring that the model benefits from as much labeled information as possible.
>
> We appreciate the reviewer for spending time reviewing our paper and offering valuable suggestions. If you have any further questions, please tell us and we are willing to address your concerns.

---

### Decision · Program_Chairs · 2025-05-01

**Decision:**

Accept (poster)

**Comment:**

The paper introduces a multilevel informed prompt tuning (MIPT) framework to enhance pretrained GNNs for molecular property prediction. The approach incorporates multi-level prompt learning to capture task-specific knowledge and a noise penalty mechanism to enhance robustness. All reviewers support acceptance, highlighting the method's novelty, thorough experimentation, and practical relevance. While some reviewers point out limitations such as missing comparisons with additional prompt-based baselines or moderate gains on specific datasets, the rebuttal effectively addressed these concerns through extended results and further analysis.